# Endogenous CO_2_ Overpressure Effect on Higher Alcohols Metabolism during Sparkling Wine Production

**DOI:** 10.3390/microorganisms11071630

**Published:** 2023-06-22

**Authors:** María del Carmen González-Jiménez, Juan Carlos Mauricio, Jaime Moreno-García, Anna Puig-Pujol, Juan Moreno, Teresa García-Martínez

**Affiliations:** 1Department of Agricultural Chemistry, Edaphology and Microbiology, Agrifood Campus of International Excellence CeiA3, University of Cordoba, Ctra. N-IV-A, km 396, 14014 Cordoba, Spain; b02gojim@uco.es (M.d.C.G.-J.); b62mogaj@uco.es (J.M.-G.); qe1movij@uco.es (J.M.); mi2gamam@uco.es (T.G.-M.); 2Department of Enological Research, Institute of Agrifood Research and Technology, Catalan Institute of Vine and Wine (IRTA, INCAVI), Plaça Àgora 2, 08720 Barcelona, Spain; anna.puig@irta.cat

**Keywords:** sparkling wine, *Saccharomyces cerevisiae*, higher alcohols, amino acids, proteins

## Abstract

Higher alcohols produced by yeast during the fermentation of sparkling wine must have the greatest impact on the smell and taste of wine. At present, the metabolic response to methanol and higher alcohols formation of *Saccharomyces cerevisiae* under endogenous CO_2_ overpressure has not been fully elucidated. In this work, a proteomics and metabolomics approach using a OFFGEL fractionator and the LTQ Orbitrap for the protein identification, followed by a metabolomic study for the detection and quantification of both higher alcohols (GC-FID and SBSE-TD-GC-MS) and amino acids (HPLC), was carried out to investigate the proteomic and metabolomic changes of *S. cerevisiae* in relation to higher alcohols formation under a CO_2_ overpressure condition in a closed bottle. The control condition was without CO_2_ overpressure in an open bottle. Methanol and six higher alcohols were detected in both conditions, and we have been able to relate to a total of 22 proteins: 15 proteins in the CO_2_ overpressure condition and 22 proteins in the control condition. As for the precursors of higher alcohols, 18 amino acids were identified in both conditions. The metabolic and proteomic profiles obtained in both conditions were different, so CO_2_ overpressure could be affecting the metabolism of higher alcohols. Furthermore, it was not possible to establish direct correlations in the condition under CO_2_ overpressure; however, in the condition without pressure it was possible to establish relationships. The data presented here can be considered as a platform that serves as a basis for the *S. cerevisiae* metabolome–proteome with the aim of understanding the behavior of yeast under conditions of second fermentation in the production of sparkling wines.

## 1. Introduction

The quality of alcoholic beverages such as wine is determined by the production of volatile aromatic compounds during fermentation [1,2]. The most important volatile aromatic compounds produced by yeast are esters, fatty acids, and higher alcohols, and their syntheses involve a highly interconnected metabolic network [3]. These compounds and their derived metabolites contribute significantly to the taste and aroma of wine [4,5,6,7,8,9]. Additionally, its amount depends on the variety of the grape, the type of yeast, the composition of the must, the fermentation temperature, winemaking practices, and the aging time [10,11,12].

Higher alcohols are quantitatively the largest group of flavor-contributing compounds in alcoholic beverages, which are present at concentrations of 10 mg/L or higher in wine [13]. Depending on the concentration in which the fusel alcohols are found, they can have a positive or negative sensory impact. Its concentrations range between 100 and 400 mg/L [4]; concentrations above 400 mg/L give a pungent solvent-like aroma to the wine, while concentrations of less than 300 mg/L are often described as imparting a desirable fruity characteristic [14,15]. Propanol, butanol, and isobutanol provide an alcoholic odor, active amyl alcohol and isoamyl alcohol have a marzipan or banana aroma, while tyrosol and 2-phenylethanol contribute with floral and honey aromas, respectively [4].

In general, the synthesis of higher alcohols takes place from the metabolism of amino acids via the Ehrlich pathway in the mitochondria and in the cytoplasm [16] or from the anabolic pathway from glucose via pyruvate [15]. Both synthesis pathways converge in the formation of an α-keto acid that can be decarboxylated and reduced to generate a higher alcohol. The Ehrlich pathway directly involves the amino acids leucine, valine, isoleucine, phenylalanine, tyrosine, and tryptophan. So, the formation of these alcohols depends on the concentration of nitrogen and amino acids [4,17,18]. The metabolic network involved in the production of these compounds during still wine fermentation has been well studied [3] but less so during second fermentation in closed bottles for sparkling wine production. The relationship between volatile compounds and the proteome of the yeast *Saccharomyces cerevisiae* is complex. In recent years, our research group has focused on the sensory and metabolic study of sparkling wines [3,19,20,21,22]. Recently, the metabolism of esters was studied, and the first metabolome–proteome relationships of *S. cerevisiae* were established during the second fermentation of sparkling wines [23]. Continuing with this purpose, the main objective of this work was to study the metabolism of higher alcohols and their relationship with the *S. cerevisiae* proteome during the second fermentation of sparkling wine production, due to the great contribution of these volatile compounds to the organoleptic properties of the wine and, therefore, to its final quality. In addition, this work has been complemented by the study of the CO_2_ overpressure that yeasts encounter during the production process and how it affects the metabolism of higher alcohols.

## 2. Materials and Methods

### 2.1. Microorganism, Grape Must, and Fermentation Conditions

In this work, the typical *S. cerevisiae* P29 strain (CECT11770) in the elaboration of sparkling wines was used. It was isolated from the designation of origin (DO) of the Penedès (Barcelona, Spain). To obtain a starter culture with a high concentration of yeast cells, a medium rich sugar was used—a pasteurized must of the Macabeo grape variety (174.9 g/L of sugar, 18.5° Bx, 3.6 g/L of total acidity and 3.43 pH) using gentle agitation of 100 rpm for 5 days and at 21 °C. When an ethanol content of 10.39% (*v*/*v*) was reached, yeast cells were used as inoculum for second fermentation. To carry out the second fermentation in the bottle using the traditional method, a base wine (Macabeo:Chardonnay (6:4), 10.21% (*v*/*v*) of ethanol, 0.3 g/L of sugar, pH 3.29, 5.4 g/L of total acidity, and 0.21 g/L of volatile acidity) previously made using traditional alcoholic fermentation was used, to which sugar (21 g/L) and 1.5 × 10^6^ yeast cells/mL were added. The second fermentation was carried out in a thermostatic chamber (Grand Cru WK/GWK 708, Liebherr, Baden, Germany) at 14 °C in bottles with a volume of 750 mL.

The bottles were divided into two groups to study the effect of CO_2_ overpressure on the metabolism of methanol and higher alcohols. For the CO_2_ overpressure condition, P (+), half of the bottles were hermetically sealed with a stopper and with a metal crown cap. The rest of the bottles were closed with a perforated shutter constituting the condition without CO_2_ overpressure, P (-), and used as a control condition.

For greater monitoring and control of the second fermentation, two sampling points were taken: at middle of fermentation (MF) when the CO_2_ overpressure in sealed bottles, measured using a pressure gauge (Mei-Heca Group, Zaragoza, Spain), reached around 3.3 atmospheres (atm) (day 8), and at the end of second fermentation (EF), when CO_2_ overpressure inside the bottles reached 6.5 atm (day 26). Residual sugars were 0.3 g/L in both cases (P (+) and P (-)). The two sampling moments were chosen based on the growth curve and metabolism of the yeasts: when yeasts are in the middle of the exponential phase of growth and express their maximum metabolism (MF samples) and when the yeasts are in the stationary phase, in a situation of maximum stress, without fermentable sugars and with the maximum pressure exerted by CO_2_ inside the bottle (EF).

### 2.2. Metabolic Analysis

#### 2.2.1. Analysis of Methanol and Higher Alcohols

Two different chromatographic methods were used to identify the largest possible number of higher alcohols. Methanol, 1-propanol, isobutanol, isoamylic alcohols, and 2-phenylethanol were detected using the direct injection method in an Agilent 6890 Series II gas chromatograph (Agilent Technologies, Palo Alto, CA, USA) equipped with a fused silica capillary column CP-WAX 57 CB (60 m long, 0.25 mm internal diameter, and 0.4 μm film thickness, from Agilent J&W, Santa Clara, CA, USA) attached to an FID detector. A solution of 1 g/L of 4-methyl-2-pentanol in pure ethanol was used as an internal standard. The chromatographic conditions are described in detail in Peinado et al. (2004) [24].

Hexanol and 2-ethyl-1-hexanol were analyzed by extraction by magnetic stir bar stirring, followed by thermal desorption and gas chromatography coupled to a mass spectrometer (SBSE-TD-GC-MS). The SBSE-TDU-GC-MS analytical platform consisted of an Agilent-7890A chromatograph, an MSD 5975 mass detector (Agilent Technologies, Wilmington, DE, USA), and the Gerstel thermal desorption unit (TDU) coupled to a CIS-4 injection system, Agilent, and an HP-5 capillary column 30 m long, 0.25 mm internal diameter and 0.25 μm film thickness. This method consisted of introducing 1 mL of sparkling wine sample into a 10 mL vial with a 12% hydroalcoholic solution in ethanol (*v*/*v*) buffered to pH 3.5, together with 0.1 mL of internal standard solution (0.446 mg/L of ethyl nonanoate in ethanol). The samples were then shaken at 1200 rpm for 100 min at 20 °C with a stirrer bar coated with polydimethylsiloxane (PDMS) (Twister) with 0.5 mm film thickness and 10 mm length, by Gerstel (GmbH, Mülheim an der Rühr, Germany). The Twister was removed from the vial, rinsed with distilled water, dried, and finally transferred to a thermal desorption unit (TDU) of Gerstel for analysis by GC/MS. The chromatographic conditions are detailed in Vararu et al. (2016) [25].

#### 2.2.2. Analysis of Amino Acids

The determination of amino acids was performed using high-performance liquid chromatography, HPLC (Agilent Technologies, mod. 1260 Infinity, Santa Clara, CA, USA), in the reverse phase by precolumn derivatization reaction with orthophthaldehyde (OPA) in the presence of mercaptoethanol. The liquid chromatograph used consisted of a 1525 pump system, a 2475 fluorescence detector, 356 nm excitation and 445 nm emission wavelength (λ), and a 717 automatic injector (Waters, Milford, Massachusetts, USA). Equipment monitoring and data acquisition and processing were performed through Waters’ Breeze program. Separations were carried out using a Nova-Pack C18 column (3.9 × 150 mm) from Waters. OPA was used as a derivatizing reagent; 750 mg of OPA was dissolved in 5 mL of methanol and 0.5 mL of 2-mercaptoethanol was added. The protocol was followed, and the chromatographic conditions used are detailed in Pripis-Nicolau et al. (2001) [26].

### 2.3. Proteomic Analysis

The yeast cells of each condition and in each time were collected by 4500× *g* for 10 min in a Routine-38 centrifuge at 4 °C. The pellet that was formed was washed twice with sterile cold distilled water. For the extraction of the proteins, the yeast cells were broken using a mechanical technique in Vibrogen Cell Mill V6 (Edmund Bühler, Bodelshausen, Germany) with glass beads of 500 μm in diameter (Sigma, St. Louis, MO, USA). These cells were resuspended in 1 mL of extraction buffer (100 mM Tris-HCl pH 8, 0.1 mM EDTA, 2 mM DTT, and 1 mM PMSF) and a cocktail of protease inhibitors provided by Roche. Then, 10% trichloroacetic acid and 4 volumes of cold acetone were added to the obtained supernatant, so that the proteins precipitated. It was left to incubate overnight at −20 °C. After the incubation period, the samples were centrifuged at 16,000× *g* for 30 min and the resulting pellet was resuspended using solubilization buffer (8 M urea, 2 M thiourea, 4% CHAPS, and 1% dithiothreitol). Later, the protein concentration was estimated using the Bradford test (1976) [27] to subsequently proceed to protein analysis. For this, 500 µg of total protein from each condition and replica was loaded into the tray of the well of the OFFGEL 3100 fractionator from Agilent Technologies (Palo Alto, CA, USA). Once the proteins were separated according to their isoelectric point, the fractions were collected from each well and their identification was carried out on an LTQ Orbitrap XL (Thermo Fisher Scientific, San Jose, CA, USA) mass spectrometer equipped with a nano-LC Ultimate 3000 system (Dionex, Germering, Germany), at the Central Support Service to Research (SCAI) of the University of Córdoba. The proteins have to be digested with trypsin beforehand. The analysis techniques described above have been published in several articles by our research team [23,28,29].

Finally, the proteins identified were quantified following the Exponentially Modified Protein Abundance Index, EmPAI, a method described by Ishihama et al. (2005) [30].

### 2.4. Statistical Analysis

All experiments were carried out in triplicate. The data was previously normalized with the square root and then scaled using the Pareto method, to avoid the differences introduced by the units of measurement [31].

The data were processed using the Statgraphics Centurion XVI.II statistical package, from STSC, Inc. (Rockville, MD, USA). The statistical tests applied were the ANOVA and Fisher’s test for the establishment of homogeneous groups (HG), with a significance level of *p* ≤ 0.05; a principal component analysis (PCA) was performed. In addition, a correlation analysis to establish significant relationships between metabolites and proteins was carried out according to the Metaboanalyst database (https://www.metaboanalyst.ca/, accessed on 30 May 2023).

## 3. Results

A total of methanol and six higher alcohols were identified in both study conditions; these were 1-propanol, isobutanol, isoamyl alcohols (2-methyl-1-butanol and 3-methyl-1-butanol), 2-phenylethanol using GC-FID, and hexanol and 2-ethyl-1-hexanol using the SBSE-TD-GC-MS technique. The data are shown in Table 1. Most of the identified higher alcohols are synthesized through the Ehrlich pathway from amino acid precursors [15], while methanol can come from the hydrolysis of pectins in the grape cell walls [32,33].

In this work, the metabolic profile obtained from the evolution of the concentration of methanol and higher alcohols was different under each condition. Under CO_2_ overpressure conditions, a decrease in the amount of methanol, 1-propanol, isobutanol, isoamyl alcohols, and 2-phenylethanol was obtained. Hexanol concentration remained constant during the second fermentation. Under the control condition, the concentrations remained constant except for propanol that increased and 1-hexanol, which decreased. The reduction in the content of 1-hexanol may have occurred due to the formation of the corresponding ester, hexyl acetate, as a product of yeast metabolism [33,34]. Tao et al. (2008) [35] found that the main higher alcohols in wine are isobutanol, phenylethanol, and isoamyl alcohol. These results obtained differ from those obtained by other authors who observe an increase in propanol, isoamyl alcohol, and isobutanol [35,36], while others obtain a decrease in these compounds [37,38]. These differences in concentrations could be due to adsorption–desorption processes in the cell walls during the aging of the wine in contact with the lees [39,40], to variations in the intracellular composition of the substrates and intermediates involved in its synthesis [41], or could be caused by the effect of CO_2_ overpressure, due to conditions of CO_2_ overpressure, P (+); the concentrations obtained decreased with respect to the control condition, P (-).

Isoamyl alcohols and 2-phenylethanol show contents, respectively, that are ten- and five-times higher than their Odor Perception Threshold, which is consistent with their classification as powerful active odorants of sparkling and still wine aromas [21]. In accordance with these authors, the isoamyl alcohols, whose aroma descriptors are wine, whiskey, ripe fruit, and nail polish, contribute in a higher extension to smell intensity than it does to wine aroma quality. Nevertheless, 2-phenyl-ethanol, with rose or honey smells, contributes pleasant notes to the wine aroma. Because of this difference, the aromatic profile of the final wine obtained could be different; total production of higher alcohols increases as concentrations of the corresponding amino acids increase. Regarding this, a preliminary sensorial analysis made by the tasters of our research team classifies these wines as a good example of the traditional sparkling cava wines from Spain. Regarding this, a preliminary organoleptic analysis made by tasters of our research team classifies these wines as representatives of the traditional sparkling wines from Spain.

With the aim of contrasting this hypothesis and relating the total production of higher alcohols to the concentrations of their corresponding amino acids, the amino acid precursors and the *S. cerevisiae* proteome were analyzed under both conditions.

A total of eighteen amino acids were identified and quantified during the second fermentation using the OPA method. The data are shown in Table 2.

Under CO_2_ overpressure conditions, an increase in the concentration of all the amino acids studied, except valine and methionine which did not show significant differences, was reported. Despite the amino acids alanine, arginine, asparagine, aspartate, glutamate, glutamine, and serine nitrogen sources being preferred by yeasts [42], an increase in the concentration of these amino acids was observed. In this condition, significant inverse correlations could be established between the amino acids identified with isoamyl alcohol, methanol, and propanol (Figure 1A). By increasing the concentration of amino acids, there is a decrease in the concentration of these alcohols. However, Belda et al. (2017) [43] reported that as the concentration of amino acids increases, that of alcohols increases.

Under the condition without CO_2_ overpressure, the amount of GABA, glutamine, asparagine, threonine, glycine, serine, isoleucine, aspartate, and tryptophan increased during the second fermentation, while the concentration of leucine and glutamate decreased (Figure 2). This release of amino acid toward the end of the fermentation and during aging is a widely documented phenomenon associated with the progressive loss of viability and later autolysis of the yeast [44]. The yeast may be using these amino acids to continue the development of other biological processes such as protein synthesis. Due to the increased concentration of alcohol in the medium, yeasts may be activating their transcription and translation machinery to try to cope with cell death and autolysis, allowing the survival of younger yeast cells by recycling the compounds released by dead yeasts [45].

In this condition, a direct and significant correlation was established between threonine and propanol. Meanwhile, inverse relationships were obtained between 2-phenylethanol and tyrosine; hexanol and GABA; glutamine, asparagine, threonine, glycine, and serine; and 2-ethyl-1-hexanol and alanine (Figure 1B).

On the other hand, alcohols were related to a total of 22 proteins involved in their metabolism: 15 proteins in the CO_2_ overpressure condition, P (+), and 22 proteins in the control condition, P (-). The data are shown in Table 3. Regarding the proteins involved in the metabolism of these alcohols, a lower number of proteins were identified under the CO_2_ overpressure condition than under the nonoverpressure condition. Under the CO_2_ overpressure condition, P (+), fourteen proteins were identified in the middle (MF) and seven proteins at the end of the fermentation (EF). Meanwhile, under nonoverpressure conditions, P (-), 22 and 14 proteins were identified in MF and EF, respectively.

In general, under the CO_2_ overpressure condition, the protein content of the identified proteins was lower with respect to the condition without CO_2_ overpressure; in addition, during the second fermentation, a decrease in the protein content was reported, except for Adh1p, Bat1p, Pdc1p, and Pdc5p. These proteins could be responsible for the increase in the concentration of higher alcohols observed at the end of the second fermentation. These proteins catalyze a different stage in the Ehrlich pathway, and they are enough to carry out the final stage in the formation of fusel alcohols in S. cerevisiae. In the catabolism of the amino acids phenylalanine, tryptophan, valine, isoleucine, and leucine, the Pdc1p, Pdc5p, or Pdc6p proteins perform the decarboxylation [46]. Furthermore, direct correlations were established between the proteins involved in the metabolism of higher alcohols, isoamyl alcohol, and methanol except for these four proteins (Figure 1A).

On the other hand, under the CO_2_ non-overpressure condition, the proteins Adh1p, Ald4p, Ald6p, Aro8p, Aro9p, and Pdc1p increased their protein content in EFP (-). On the contrary, the rest of the proteins decreased their content. As in the previous condition, proteins belonging to each stage of the formation of fusel alcohols have been identified. Because no increase in the concentration of higher alcohols was observed, as expected, the formation of fusel acids could take place, since the Ald4p and Ald6p proteins were identified. Ald4p and Ald6p are responsible for reducing the aldehyde of fusel to a fusel acid. These proteins were directly correlated with propanol and inversely with hexanol (Figure 1B). This could suggest the possible formation of propanol via the Ehrlich pathway from threonine.

It should be noted that in both study conditions the Adh2 and Thi3p proteins were not identified, which have been related to a reduced production of acetaldehyde or fusel alcohols, respectively [47]. These data may also be indicative of the reduction in alcohols found in our work.

Generally, the amine group of most amino acids enters the central nitrogen metabolism through transamination, thereby releasing the carbon skeleton as a 2-oxoacid. These compounds can directly enter the central carbon metabolism, while the 2-oxoacids from aromatic and branched-chain amino acids can decarboxylate through the so-called Ehrlich pathway and produce the corresponding aldehyde [48]. Depending on the availability of oxygen, the aldehyde can be oxidized to a carboxylic acid, called fusel acid, or reduced to a fusel alcohol [49]. Both acids and fusel alcohols and their derivatives contribute greatly to the flavor of the wine.

To carry out the analysis of principal components (PCA), the number of variables was reduced according to the multiple analysis of homogeneous groups, staying with those that presented HG ≥ 3. The resulting PCA (Figure 3) explains 96.84% of the total variance. Component 1 (PC 1) explains 43.99% and helps to discriminate the sampling times, that is, half of the fermentation with negative values and the end of the second fermentation in positive values. This component is strongly correlated with Adh5p (−0.262423) for half the fermentation and asparagine (0.266257), tryptophan (0.26102), and glutamine (0.259309) for the end of fermentation. These have been surrounded by green color. Component 2 (PC 2), in yellow color, explains 29.30% of the variability and distinguishes, with positive values, the end of the fermentation with pressure and, in negative values, the end of the fermentation without pressure; four proteins contribute to this component, Ald2p (−0.326691), Ald4p (−0.319366), Ald6p (−0.319599), and Aro8p (−0.302044), for the condition without CO_2_ overpressure and the amino acid leucine (0.292128) and Pdc5p (0.276056) for the CO_2_ overpressure condition at the end of the second fermentation. Meanwhile 1-propanol (0.311078), Adh7p (−0.312687), Ald5p (−0.317511), and Bat2p (−0.317433) are related to component 3, which explains 23.55% of the variance. This component differentiates half of the fermentation with pressure in positive values and the fermentation half without CO_2_ overpressure with negative values. This component has been represented with the red color.

## 4. Conclusions

This work is focused on establishing correlations between the metabolite of higher alcohols and the proteome of *S. cerevisiae* during the second fermentation in the production of Spanish sparkling wine (cava). We investigated the effect of CO_2_ overpressure in a typical yeast to produce sparkling wines, through the study of the metabolism of higher alcohols, due to its influence on the organoleptic properties of sparkling wines.

In general, under the CO_2_ overpressure condition, the concentration of the higher alcohols decreased during the second fermentation, while in the condition without pressure the concentration remained constant for most of these compounds. In protein terms, in the pressure condition, a smaller number of proteins were identified than in the non-pressure condition.

It was not possible to establish significant direct correlations between amino acids, alcohols, and proteins in the condition under CO_2_ overpressure; on the contrary in the condition without overpressure, propanol may be synthesized through threonine via the Ehrlich pathway.

Regarding the CO_2_ overpressure to which the yeasts are subjected, it could be concluded that the CO_2_ overpressure affects the metabolism of the alcohols in the yeast.

The wines did not show sensorial defects and were positively evaluated by the researchers.

The results presented in this work are the beginning in the search for metabolome-proteome relationships of the yeast *S. cerevisiae* during the production of Spanish sparkling wine (cava). However, additional research involving transcriptomic and genetic analyzes would be necessary to reach more robust conclusions.

## Figures and Tables

**Figure 1 microorganisms-11-01630-f001:**
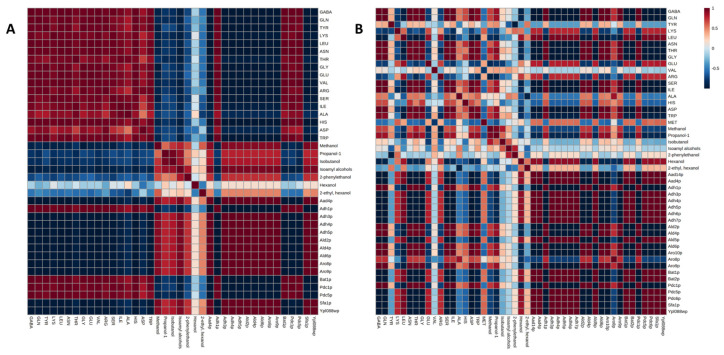
Matrix resulting from the analysis of correlations made for the condition with CO_2_ overpressure (**A**) and condition without CO_2_ overpressure (**B**) with a level of significance of 95%.

**Figure 2 microorganisms-11-01630-f002:**
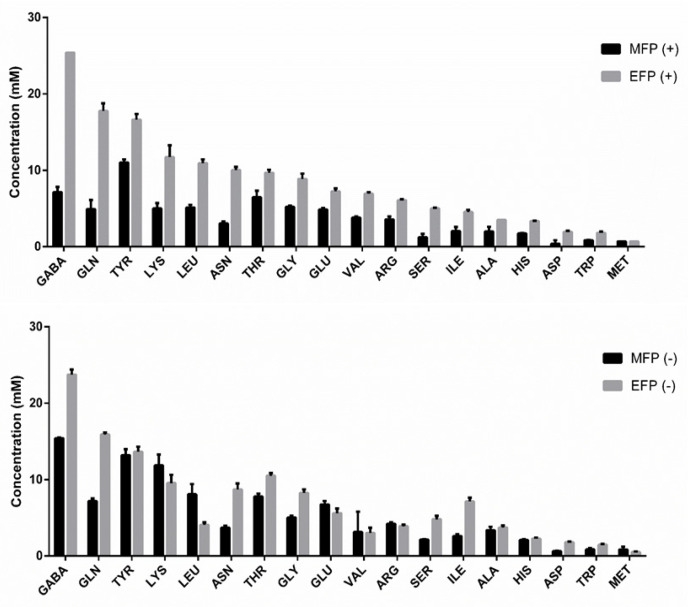
Determination of the concentration (mM) of amino acids using high-performance liquid chromatography, HPLC, in the reverse phase via the precolumn derivatization reaction with orthophthaldehyde (OPA), under the CO_2_ overpressure condition (above) and without CO_2_ overpressure condition (below). MFP (+): middle of fermentation with pressure; EFP (+): end of pressure fermentation; MFP (-): middle of fermentation without pressure; EFP (-): end of fermentation without pressure.

**Figure 3 microorganisms-11-01630-f003:**
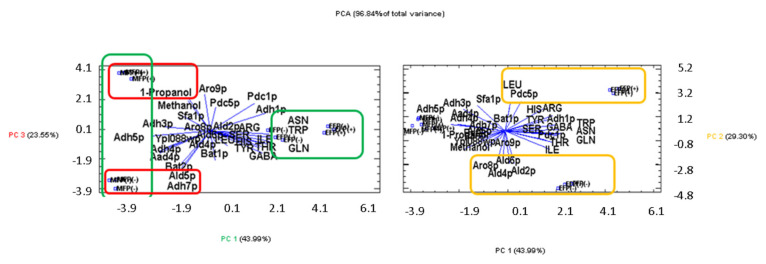
Representation in the defined plane of three main components (PCA). MFP (+): middle of fermentation with pressure; MFP (-): middle of fermentation without pressure; EFP (+): end of fermentation with pressure; EFP (-): end of fermentation without pressure.

**Table 1 microorganisms-11-01630-t001:** Determination of the concentration (mg/L) of methanol and higher alcohols in four conditions (MFP (+), MFP (-), EFP (+), EFP (-)) using the GC-FID and SBSE-TD-GC-MS methods. Odor Perception Threshold (OPT).

	Concentration (mg/L)				
	OPT (mg/L)	MFP (+)	EFP (+)	MFP (-)	EFP (-)
Methanol	668	30.85 ^c^ ± 1.11	26.80 ^a^ ± 1.60	26.82 ^ab^ ± 1.39	29.10 ^bc^ ± 1.81
1-Propanol	830	15.01 ^c^ ± 1.20	12.89 ^ab^ ± 0.49	11.97 ^a^ ± 0.62	14.12 ^bc^ ± 0.51
Isobutanol	40	21.03 ^b^ ± 0.04	18.96 ^a^ ± 0.96	19.69 ^a^ ± 0.33	19.41 ^a^ ± 0.42
Isoamylic Alcohols	30	296.33 ^b^ ± 3.29	260.97 ^a^ ± 10.01	288.52 ^b^ ± 15.98	295.61 ^b^ ± 0.25
2-phenylethanol	10	58.06 ^b^ ± 2.99	51.11 ^a^ ± 3.09	52.99 ^ab^ ± 2.00	52.28 ^ab^ ± 1.14
Hexanol	2.5	0.69 ^a^ ± 0.72	0.68 ^a^ ± 0.42	0.82 ^b^ ± 0.33	0.69 ^a^ ± 0.40
2-ethyl-1-hexanol	8	0.91 ^ns^ ± 0.27	0.72 ^ns^ ± 0.17	0.60 ^ns^ ± 0.17	0.49 ^ns^ ± 0.13

Different letters (a–c) in the same row are significantly different (*p* ≤ 0.05) according to Fisher’s test; ns = non-significant. MFP (+): middle of fermentation with pressure; MFP (-): middle of fermentation without pressure; EFP (+): end of fermentation with pressure; EFP (-): end of fermentation without pressure.

**Table 2 microorganisms-11-01630-t002:** Determination of the concentration (mM) of the different amino acids via the OPA method using HPLC.

	Concentration (mM)			
	MFP (+)	EFP (+)	MFP (-)	EFP (-)
GABA	7.1 ^a^ ± 0.7	25.4 ^d^ ± 0.0	15.4 ^b^ ± 0.1	23.7 ^c^ ± 0.7
GLN	5.0 ^a^ ± 1.2	17.8 ^c^ ± 1.0	7.2 ^b^ ± 0.3	15.9 ^c^ ± 0.3
TYR	11.0 ^a^ ± 0.4	16.6 ^c^ ± 0.8	13.2 ^b^ ± 0.8	13.6 ^b^ ± 0.7
CYS	5.0 ^a^ ± 0.7	11.8 ^b^ ± 1.5	11.9 ^b^ ± 1.4	9.5 ^b^ ± 1.1
LEU	5.1 ^a^ ± 0.4	11.0 ^c^ ± 0.5	8.0 ^b^ ± 1.3	4.1 ^a^ ± 0.3
ASN	3.0 ^a^ ± 0.2	10.0 ^d^ ± 0.4	3.7 ^b^ ± 0.3	8.7 ^c^ ± 0.8
THR	6.5 ^a^ ± 0.9	9.7 ^c^ ± 0.4	7.8 ^b^ ± 0.4	10.5 ^c^ ± 0.3
GLY	5.3 ^a^ ± 0.1	8.9 ^b^ ± 0.7	5.0 ^a^ ± 0.2	8.3 ^b^ ± 0.4
GLU	4.8 ^b^ ± 0.2	7.2 ^a^ ± 0.4	6.7 ^b^ ± 0.5	5.6 ^a^ ± 0.6
VAL	3.8 ^ns^ ± 0.2	6.9 ^ns^ ± 0.2	3.1 ^ns^ ± 2.6	3.0 ^ns^ ± 0.7
ARG	3.6 ^a^ ± 0.4	6.1 ^b^ ± 0.1	4.2 ^a^ ± 0.1	3.9 ^a^ ± 0.2
SER	1.2 ^b^ ± 0.5	5.0 ^c^ ± 0.1	2.1 ^a^ ± 0.1	4.8 ^c^ ± 0.4
ILE	2.0 ^a^ ± 0.6	4.5 ^b^ ± 0.3	2.6 ^a^ ± 0.3	7.1 ^c^ ± 0.5
ALA	2.0 ^a^ ± 0.6	3.5 ^b^ ± 0.0	3.4 ^b^ ± 0.5	3.7 ^b^ ± 0.2
HIS	1.7 ^a^ ± 0.1	3.3 ^c^ ± 0.1	2.1 ^b^ ± 0.1	2.3 ^b^ ± 0.2
ASP	0.4 ^a^ ± 0.4	1.9 ^b^ ± 0.2	0.6 ^a^ ± 0.1	1.8 ^b^ ± 0.1
TRP	0.8 ^a^ ± 0.0	1.8 ^c^ ± 0.1	0.9 ^a^ ± 0.2	1.5 ^b^ ± 0.1
MET	0.7 ^ns^ ± 0.0	0.7 ^ns^ ± 0.0	0.8 ^ns^ ± 0.3	0.5 ^ns^ ± 0.1

n.d. = not detected. Different letters (a–d) in the same row are significantly different (*p* ≤ 0.05) according to Fisher’s test. MFP (+): middle of fermentation with pressure; MFP (-): middle of fermentation without pressure; EFP (+): end of fermentation with pressure; EFP (-): end of fermentation without pressure; ns = non-significant.

**Table 3 microorganisms-11-01630-t003:** Determination of protein content (mol%) using the abundance index (EmPAI) of the proteins identified in each condition (MFP (+), MFP (-), EFP (+), and EFP (-)) related to the synthesis of fusel alcohols using the Ehrlich pathway.

Protein Name	Protein Content (mol%)			
	MFP (+)	EFP (+)	MFP (-)	EFP (-)
**^1^ AAD14P**	n.d. ^a^	n.d. ^a^	0.0355 ^b^ ± 0.0004	n.d. ^a^
**^1^ AAD4P**	0.0149 ^b^ ± 0.0001	n.d. ^a^	0.0503 ^c^ ± 0.0005	n.d. ^a^
**^1^ ADH1P**	0.424 ^a^ ± 0.004	0.841 ^c^ ± 0.008	0.200 ^b^ ± 0.002	0.539 ^d^ ± 0.005
**^1^ ADH3P**	0.122 ^c^ ± 0.001	0.078 ^a^ ± 0.001	0.121 ^c^ ± 0.001	0.0356 ^b^ ± 0.0004
**^1^ ADH4P**	0.0195 ^b^ ± 0.0002	n.d. ^a^	0.0675 ^c^ ± 0.0007	n.d. ^a^
**^1^ ADH5P**	0.0404 ^b^ ± 0.0004	n.d. ^a^	0.0578 ^c^ ± 0.0004	n.d. ^a^
**^1^ ADH6P**	n.d. ^a^	n.d. ^a^	0.0567 ^b^ ± 0.0006	n.d. ^a^
**^1^ ADH7P**	n.d. ^a^	n.d. ^a^	0.120 ^b^ ± 0.001	0.0187 ^c^ ± 0.0002
**^2^ ALD2P**	0.0102 ^c^ ± 0.0001	n.d. ^a^	0.0091 ^b^ ± 0.0001	0.0399 ^d^ ± 0.0004
**^2^ ALD4P**	0.0365 ^b^ ± 0.0004	0.0120 ^a^ ± 0.0001	0.0519 ^c^ ± 0.0005	0.109 ^d^ ± 0.001
**^2^ ALD5P**	n.d. ^a^	n.d. ^a^	0.0280 ^c^ ± 0.0003	0.0110 ^b^ ± 0.0001
**^2^ ALD6P**	0.0295 ^d^ ± 0.0003	n.d. ^a^	0.0768 ^c^ ± 0.0008	0.370 ^b^ ± 0.004
**^3^ ARO10P**	n.d. ^a^	n.d. ^a^	0.00389 ^b^ ± 0.00004	n.d. ^a^
**^4^ ARO8P**	0.0317 ^ab^ ± 0.0003	n.d. ^a^	0.111 ^b^ ± 0.001	0.211 ^c^ ± 0.002
**^4^ ARO9P**	0.0280 ^d^ ± 0.0003	n.d. ^a^	0.0061 ^b^ ± 0.0001	0.0176 ^c^ ± 0.0002
**^4^ BAT1P**	n.d. ^a^	0.0930 ^b^ ± 0.0009	0.103 ^c^ ± 0.001	n.d. ^a^
**^4^ BAT2P**	n.d. ^a^	n.d. ^a^	0.0496 ^c^ ± 0.0005	0.0187 ^b^ ± 0.0002
**^3^ PDC1P**	0.397 ^b^ ± 0.004	0.430 ^c^ ± 0.004	0.246 ^a^ ± 0.003	0.453 ^d^ ± 0.005
**^3^ PDC5P**	0.377 ^c^ ± 0.004	0.408 ^d^ ± 0.004	0.253 ^b^ ± 0.003	0.197 ^a^ ± 0.0002
**^3^ PDC6P**	n.d. ^a^	n.d. ^a^	0.0488 ^b^ ± 0.0005	n.d. ^a^
**^1^ SFA1P**	0.0913 ^c^ ± 0.0009	0.0800 ^b^ ± 0.0008	0.0793 ^b^ ± 0.0008	0.0556 ^a^ ± 0.0006
**^1^ YPL088WP**	0.0234 ^b^ ± 0.0002	n.d. ^a^	0.0443 ^d^ ± 0.0004	0.0298 ^c^ ± 0.0003

n.d. = not detected, for statistics it has been replaced by half the minimum value. Different letters (a–d) in the same row are significantly different (*p* ≤ 0.05) according to Fisher’s test. The superscripts indicate the stage of the route in which they are involved: ^1^ oxidation, ^2^ reduction, ^3^ decarboxilation and ^4^ transamination. MFP (+): middle of fermentation with pressure; MFP (-): middle of fermentation without pressure; EFP (+): end of fermentation with pressure; EFP (-): end of fermentation without pressure.

## Data Availability

The data are available upon request to the authors.

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
