# Peer review of "Endogenous CO2 Overpressure Effect on Higher Alcohols Metabolism during Sparkling Wine Production"

_microorganisms, 2023, doi:10.3390/microorganisms11071630_

Round 1

Reviewer 1 Report

The work conducted is highly interesting. Exploring the impact of pressure on yeast metabolism represents an innovative research angle. The quantity and quality of the analyses performed are impressive.

However, my only critique would be that the authors might have become somewhat lost amidst the extensive number of analyses. I find the study lacking in its discussion of technical aspects, particularly concerning sensory aspects. Indeed, the literature review clearly indicates that higher alcohols play a role in the sensory quality of the product. Unfortunately, I could not find any discussion on this topic in the conclusion.

Additionally, it would be advantageous to include information about the pressure attained in the overpressure modality (presence of residual sugars?).

line 86: sucrose and 1.5x106 cells/mL were added

Author Response

The work conducted is highly interesting. Exploring the impact of pressure on yeast metabolism represents an innovative research angle. The quantity and quality of the analyses performed are impressive.

However, my only critique would be that the authors might have become somewhat lost amidst the extensive number of analyses. I find the study lacking in its discussion of technical aspects, particularly concerning sensory aspects. Indeed, the literature review clearly indicates that higher alcohols play a role in the sensory quality of the product. Unfortunately, I could not find any discussion on this topic in the conclusion.

Thank you very much for your appreciation of the sensory analysis, this has been taken into account in the new manuscript.

Additionally, it would be advantageous to include information about the pressure attained in the overpressure modality (presence of residual sugars?).

This comment has been added in the Materials and Methods section.

line 86: sucrose and 1.5x106 cells/mL were added

  1. Corrected in the text.

Reviewer 2 Report

The work is interesting and concerns an important aspect of the quality and technology of sparkling wines. The existing dependencies and possible probable mechanisms are logically described and discussed. The conclusions are not very precise, but this is due to ambiguous research results.

Sugestions:

Line 78 For the growth of the yeast cells, these were incubated for 5 days and at 21 oC in a pasteurized must of the Macabeo grape variety. …. The second fermentation was conduct in bottles with a standardized commercial base wine – why two different media? the second fermentation takes place in the same wine (need to be explain in the methodology text)

Line 92: halfway through the second fermentation (MF) and at the end of the second fermentation (EF) – give specific parameters for sampling, halfway say nothing

References could be more up-to-date, suggesting to review and include more actual papers (last 3 years).

Author Response

Comments and Suggestions for Authors

The work is interesting and concerns an important aspect of the quality and technology of sparkling wines. The existing dependencies and possible probable mechanisms are logically described and discussed. The conclusions are not very precise, but this is due to ambiguous research results.

Sugestions:

Line 78 For the growth of the yeast cells, these were incubated for 5 days and at 21 oC in a pasteurized must of the Macabeo grape variety. …. The second fermentation was conduct in bottles with a standardized commercial base wine – why two different media? the second fermentation takes place in the same wine (need to be explain in the methodology text).

The following paragraph has been included in the new text for clarification:

In order to obtain a starter culture with a high concentration of yeast cells, a medium rich in sugars was used: a pasteurized must of the Macabeo grape variety (174.9 g/L of sugar, 18.5oBx, 3.6 g/L of total acidity and 3.43 pH) using gentle agitation of 100 rpm. When an ethanol content of 10.39% (v / v) was reached, yeast cells were used as inoculum for second fermentation. To carry out the second fermentation in the bottle using the traditional method, a base wine (Macabeo:Chardonnay (6:4), 10.21% (v/v) of ethanol, 0.3 g/L of sugar, pH 3.29, 5.4 g/L of total acidity and 0.21 g/L of volatile acidity) previously made by traditional alcoholic fermentation was used, to which sugar (21 g/L) and 1.5×106 yeast cells/mL were added.

Line 92: halfway through the second fermentation (MF) and at the end of the second fermentation (EF) – give specific parameters for sampling, halfway say nothing

The following paragraph has been included in the new text for clarification:

For greater monitoring and control of the second fermentation, two sampling points were taken: at middle of fermentation (MF) when the CO2 overpressure in sealed bottles, measured by a pressure gauge (Mei-Heca Group, Spain), reached around 3.3 atmospheres (atm) (day 8), and at the end of second fermentation (EF) when CO2 overpressure inside the bottles reached 6.5 atm (day 26). Residual sugars were 0.3 g/L in both cases (P (+) and P (-)).

References could be more up-to-date, suggesting to review and include more actual papers (last 3 years).

Current documents have been included in the new manuscript.

Reviewer 3 Report

The study could be of interest but as the authors clearly specified in the conclusions, the aim of the study was not reached. In my opinion, this study needs important improvements before being accepted and taken into consideration by the audience. 

Please give more details about thermostatic chamber (model, producer, country) -line 85

Pay attention to spelling and punctuation (eg. line 88 - period is missing). Carefully check the entire document.

Clearly specify in section 2.1 the moments of sampling - day X. Why did you choose only two moments for sampling the wine? 

Complete the description of the methods with all the equipment details (model, producers, country).

Line 163 - methanol is not a higher alcohol. Please correct this within the entire manuscript (discussion, tables, etc.).

Adjust your discussion section to avoid misunderstanding the content. Line 167 - give examples of the methane present in the grapes (as a raw material for wine production). I think you missed out on the point here. Please correct this part. 

Do not use capital letters when discussing about amino acids - lines 221-222.

Conclusions should refer more to the impact of these findings. Have you thought that you were unable to provide a correlation because the sample number was too low? 

My comments were specified above.

Author Response

Comments and Suggestions for Authors

The study could be of interest but as the authors clearly specified in the conclusions, the aim of the study was not reached. In my opinion, this study needs important improvements before being accepted and taken into consideration by the audience.

Please give more details about thermostatic chamber (model, producer, country) -line 85

The following paragraph has been included in the new text:

The second fermentation was carried out in a thermostatic chamber (Grand Cru WK/GWK 708, Liebherr, Germany) at 14 oC in bottles with a volume of 750 mL.

Pay attention to spelling and punctuation (eg. line 88 - period is missing). Carefully check the entire document.

OK corrected

Clearly specify in section 2.1 the moments of sampling - day X. Why did you choose only two moments for sampling the wine?

This has been included in the new manuscript

The two sampling moments were chosen based on the growth curve and metabolism of the yeasts: when yeasts are in the middle of the exponential phase of growth and express their maximum metabolism (MF samples) and when the yeasts are in the stationary phase, in a situation of maximum stress, without fermentable sugars and with the maximum pressure exerted by CO2 inside the bottle (EF).

Complete the description of the methods with all the equipment details (model, producers, country).

This has been included in the new manuscript

Line 163 - methanol is not a higher alcohol. Please correct this within the entire manuscript (discussion, tables, etc.).

This has been included in the new manuscript

Adjust your discussion section to avoid misunderstanding the content. Line 167 - give examples of the methane present in the grapes (as a raw material for wine production). I think you missed out on the point here. Please correct this part.

This has been corrected.

Yeasts form a small amount of methanol during alcoholic fermentation, but most of the methanol comes from the hydrolysis of pectins.

Do not use capital letters when discussing about amino acids - lines 221-222.

This has been corrected.

Conclusions should refer more to the impact of these findings. Have you thought that you were unable to provide a correlation because the sample number was too low?

You are right. But this research is about establishing new study strategies to advance our understanding of the relationship between the metabolome and the proteome, but more studies are still needed.

Round 2

Reviewer 3 Report

In my opinion, the manuscript still has to be improved and resubmitted for its reevaluation. 

In my opinion, the manuscript still has to be improved and resubmitted for its reevaluation.